# A Model for Estimating the Vegetation Cover in the High-Altitude Wetlands of the Andes (HAWA)

**Jorge Soto [1], Celián Román-Figueroa [1] and Manuel Paneque [2,\*]**

[1]   Agroenergía Ingeniería Genética S.A. Inc., Almirante Lynch 1179, San Miguel, Santiago 8920033, Chile; jsoto@bionostra.com (J.S.); croman@bionostra.com (C.R.-F.)

[2]   Department of Environmental Sciences and Natural Resources, Faculty of Agricultural Sciences, University of Chile, Santa Rosa 11315, La Pintana, Santiago 8820808, Chile

\*   Correspondence: mpaneque@uchile.cl; Tel.: +56-2-2978-5863

**Abstract:** The natural salt meadows of Tilopozo in the hyperarid, Atacama Desert of northern Chile, which are located at approximately 2800 m above sea level, are under pressure from industrial activity, and cultivation and grazing by local communities. In this research, the land surface covered by salt meadow vegetation was estimated from normalized difference vegetation indices (NDVI) derived from Landsat Thematic Mapper (TM), Enhanced Thematic Mapper (ETM+) and Operational Land Imager (OLI) data from 1985 to 2016. The vegetated area of the Tilopozo salt meadows decreased by 34 ha over the 32-year period studied. Multiple regression models of the area covered by vegetation and climate data and groundwater depths were derived on an annual basis, as well as for both the dry and wet seasons and had $R^2$ values of 83.0%, 72.8% and 92.4% respectively between the vegetated areas modeled and those estimated from remotely sensed data. These models are potentially useful tools for studies into the conservation of the Tilopozo salt meadows, as they provide relevant information on the state of vegetation and enable changes in vegetation in response to fluctuations in climate parameters and groundwater depths to be predicted.

**Keywords:** high altitude wetlands of the Andes (HAWA); land surface changes prediction models; conservation wetlands; model for vegetation cover estimation; Atacama Desert

## 1. Introduction

The importance of wetlands is greater in places where water resources are scarce, e.g., the arid and semi-arid ecosystems of South America [1,2]. In the Altiplano regions of South America, there are wetlands at high altitudes, which are located at the maximum altitude for vegetation growth and at less than the zero isotherms [1]. In these locations, natural salt meadows can be found [1,3]. These salt meadows are characterized by stagnant and saline waters [4], and have a water table that is either at the ground surface or a few meters below this. For example, the depth of the water table in the Tilopozo salt meadows is found between 0 and 3.65 m, with an average depth if 1.82 m [5,6].

The salt meadows of Tilopozo are located to the south of the Salar de Atacama, Chile [7], and the presence of the ecosystem depends on the discharge of the flow from the Monturaqui-Negrillar-Tilopozo aquifer (MNT-aquifer), which has an extension of 60 km [8]. Since 1986, water has been extracted from the MNT-aquifer for industrial purposes (metallic and non-metallic mining), and this has an established monitoring plan through a water monitoring network for the conservation of this wetland [9].

A series of studies have been undertaken to model the influences of different factors on changes in vegetation cover [10]. Among them, Hyandye et al. [11] determined that variations in precipitation, slope aspect, road network density generate changes in land use. Los et al. [12] and Hyandye et al. [11] highlighted that the analysis of satellite images and statistical models together constitutes a powerful

combination to evaluate and model change processes and their underlying causes. Another study was carried out by Chahouki and Chahouki [13], who used a digital elevation model (DEM) and statistical tools to generate a predictive map of rangelands in an arid context, obtaining a projection of vegetation distributions according to species, which was based on soil variables estimated from a DEM.

In the context of this research, the normalized difference vegetation index (NDVI) has been used to detect vegetation cover in wetlands and determine the extent of wetland areas [14]. Various studies have been developed in arid and hyper-arid zones, in which correlations between vegetation cover and rainfall have been obtained, with results that allow vegetation cover to be predicted and that have applications for the conservation or management of pastures and wetlands [15,16]. In this sense, Mwita et al. [17] mapped small wetlands using NDVI. Dong et al. [18] developed a wetland map based on the use of NDVI using Landsat images. Recently, Qu et al. [19] analyzed trends in NDVI between 1982 and 2011 for wetlands in China. They were able to correlate them with climatic conditions and indicators of human activity and developed a multiple regression model that determined the response of vegetation to climatic factors. In the context of this research, NDVI has been used to detect vegetation cover in wetlands and determine the extent of wetland areas [14]. Different authors have highlighted that NDVI is one of the most important indexes for evaluating the state of vegetation, because it is correlated with photosynthesis and primary production [20–22].

Today, there are many methods for monitoring the state of conservation of the wetlands through remote sensing [5,23]. Remote sensing allows evaluation of land use changes and variations in vegetation cover [5,20,23,24] and changes in the wetland ecosystems because of hydrological variations [23,25]. Multitemporal analysis of satellite images is usually used to identify and estimate the changes in the surface of the cover with vegetation [5,23]. Vegetation and land use change in wetland areas may also be due to abrupt changes produced by floods, fires, or human interventions [20,26]. Studies, such as those of Li et al. [20], have evaluated the dynamics and loss of marshes and meadows vegetation in wetland due to hydrological variations using NDVI data derived from Landsat Thematic Mapper (TM), Enhanced Thematic Mapper (ETM+) and Operational Land Imager (OLI) sensors, and generated accuracies up to 91%.

Consequently, the present study sought to determine changes in vegetation cover of the slat meadows of Tilopozo between 1985 and 2016, using the NDVI index estimated from Landsat TM and OLI images, and to design predictive models, for the surface covered with vegetation, on the basis of meteorological information and data from a groundwater monitoring network.

## 2. Methodology

### 2.1. Study Area

The study area corresponded to the salt meadows of Tilopozo, located to the south of the Salar de Atacama, San Pedro de Atacama, Antofagasta Region, Chile (23°46′42″S; 68°14′39″W; 2800 m above sea level; Figure 1).

The salt meadows of Tilopozo are characterized by hydrohumedal and hygrohumedal structures that are influenced by the spatial distribution of confined and unconfined groundwater. Its feeding was freatogenic of discharge, given its dependence on the waters coming from the discharge of the MNT-aquifer [27,28]. This leads to phreatically influenced azonal vegetation, with transitional hygrophilic species, such as *Distichlis spicata*, *Lycium humile* and *Tessaria absinthioides*, as well as strict hygrophilic species, such as *Baccharis juncea*, *Juncus balticus* and *Schoenoplectus americanus* (*S. americanus*) [29,30], which are all perennials. Differences in soil moisture and groundwater levels lead to surface variations, mainly in transitional hygrophilic species. The wetland contracted and expanded according to variations in annual rainfall during the time period analyzed; it was noted to expand in wet years (e.g., during the year with the highest wet season rainfall, 2002) and contract in dry years (e.g., 1991, Figures 2 and 3).

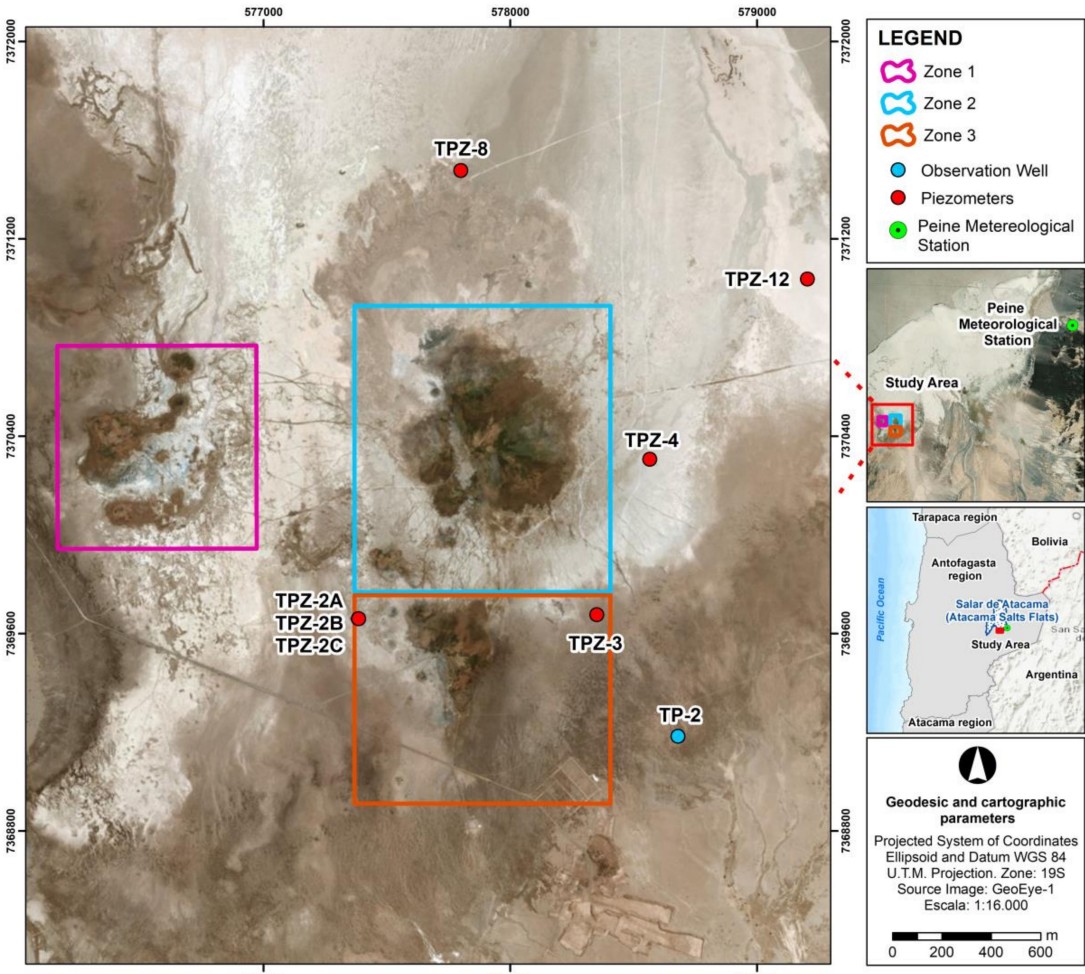

**Figure 1.** Position of the salt meadows zones of Tilopozo associates to three upwelling of water (1–3), located to the south of the Salar de Atacama.

In meteorological terms, the salt meadows of Tilopozo experience high thermal oscillations and low levels of precipitation, with extreme events in the wet season [31]. Based on records from 1975 to 2016 of the Peine meteorological station of the General Directorate of Water (DGA), average monthly precipitation was 1.67 ± 1.65 mm, most of which occurred January and March, with a maximum of 14.57 ± 16.43 mm [31]. The lowest monthly precipitation totals (0.00 ± 0.00 mm) occur between September and November. Meanwhile, the historical average annual temperature was 16.51 ± 3.31 °C. The coldest month is July, with a minimum of −0.72 ± 1.63 °C, while the warmest is December, with a maximum of 31.82 ± 1.64 °C (Figure 3). The monthly average evapotranspiration was 163.63 mm, with minimum values in the winter months of June and July, and summer peaks in December and January.

*2.2. Vegetation Cover in the Tilopozo Salt Meadows between 1985 and 2016*

The surface covered by vegetation in the salt meadows of Tilopozo was estimated using two Landsat TM or OLI images per year (dry season—September, and wet season—March), for the period 1985–2016. Images with less cloud cover were selected to avoid errors in the estimate (Table 1), then radiometric [32] and atmospheric [33] corrections were conducted. The salt meadows of Tilopozo were easily recognized using the normalized differential vegetation index (NDVI). The threshold value of NDVI for the vegetation was determined as 0.13 [21,34,35]. The use of this NDVI threshold is supported by research in other hyper-arid areas in Chile [36,37], and was complemented by visual analysis of true colour composites. This allowed the surfaces covered by vegetation to be identified for the wet and dry season images, and for the 'annual images', i.e., the averages of the wet and dry season images.

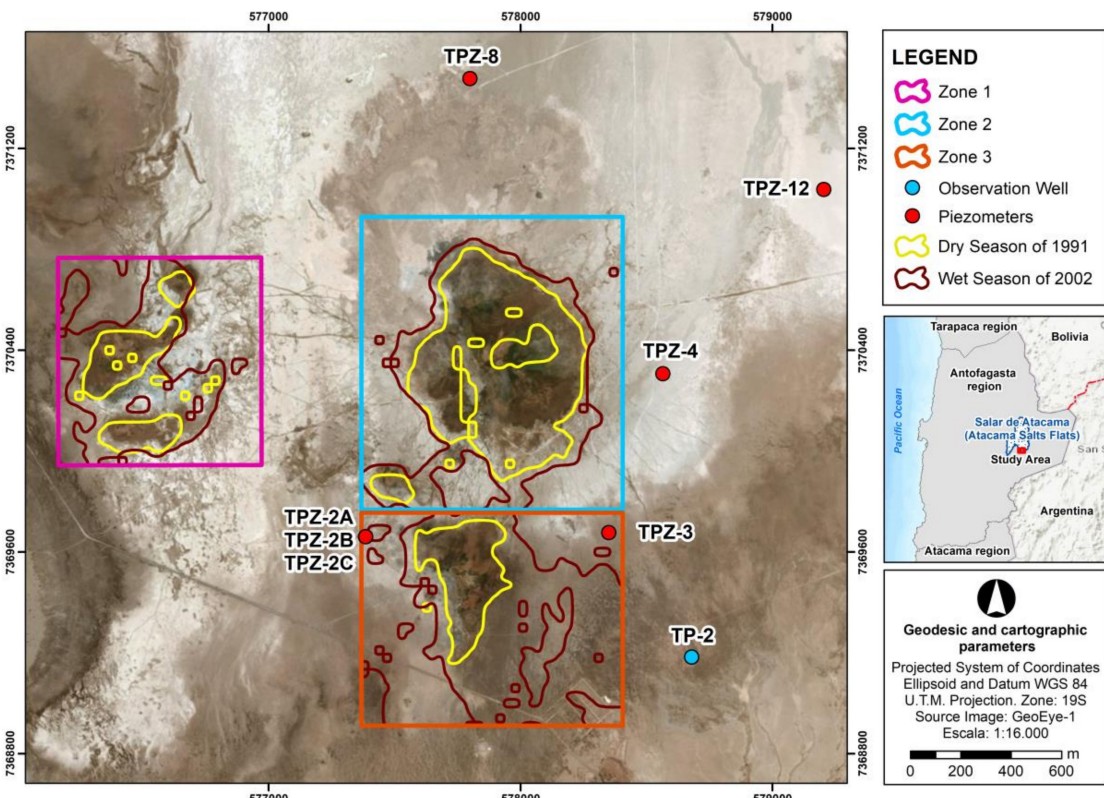

**Figure 2.** Maximum and minimum extent of vegetation cover in the Tilopozo salt meadows in the period studied.

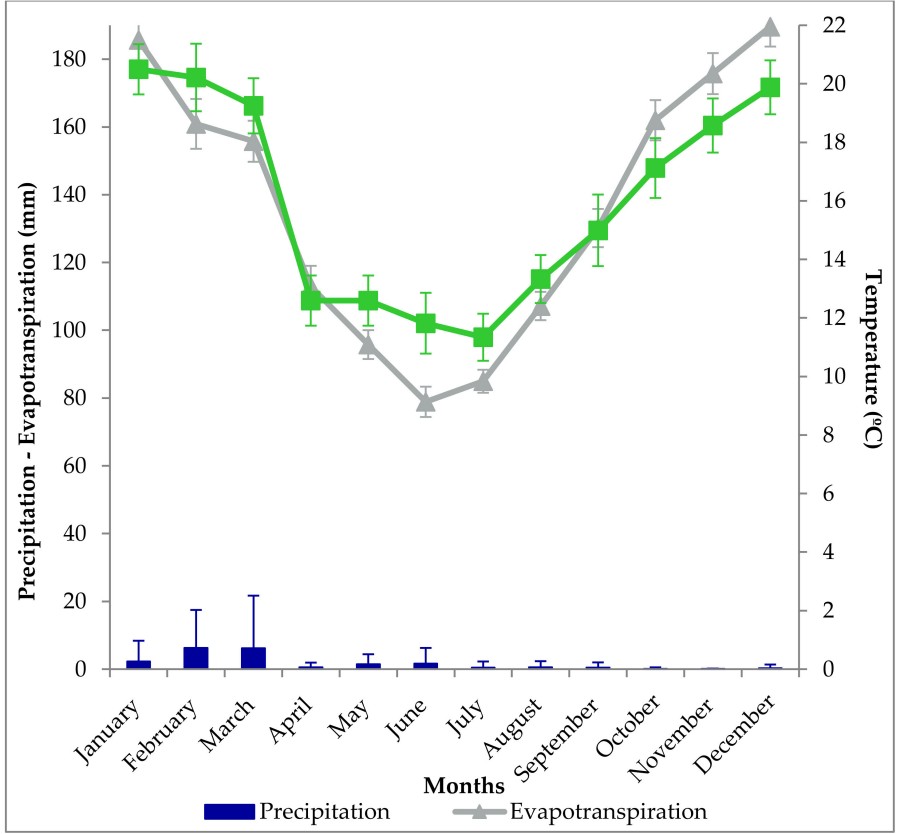

**Figure 3.** Average monthly precipitation (mm), evapotranspiration (mm) and temperature (°C) for the period 1975–2016 for Peine meteorological station.

**Table 1.** Landsat Thematic Mapper (TM) and Operational Land Imager (OLI) images used in the research.

| ID | Date | Discharge Name | Cloud | Sensor | ID | Date | Discharge Name | Cloud | Sensor |
|----|------|----------------|-------|--------|----|------|----------------|-------|--------|
| 1 | 30 March 1985 | LT52330761985089AAA03 | 10% | L5 TM | 31 | 08 April 2000 | LT52330762000099CUB00 | 12% | L5 TM |
| 2 | 05 August 1985 | LT52330761985217XXX04 | 0% | L5 TM | 32 | 23 September 2000 | LT52330762000259CUB02 | 18% | L5 TM |
| 3 | 02 April 1986 | LT52330761986092AAA08 | 0% | L5 TM | 33 | 10 March 2001 | LT52330762001069COA00 | 35% | L5 TM |
| 4 | 09 September 1986 | LT52330761986252AAA03 | 0% | L5 TM | 34 | 18 September 2001 | LT52330762001261CUB00 | 19% | L5 TM |
| 5 | 04 March 1987 | LT52330761987063AAA07 | 0% | L5 TM | 35 | 29 March 2002 | LT52330762002088COA00 | 0% | L5 TM |
| 6 | 30 October 1987 | LT52330761987303CUB00 | 1% | L5 TM | 36 | 08 September 2003 | LT52330762003251CUB00 | 31% | L5 TM |
| 7 | 06 March 1988 | LT52330761988066CUB00 | 0% | L5 TM | 37 | 18 March 2004 | LT52330762004078CUB00 | 0% | L5 TM |
| 8 | 30 September 1988 | LT52330761988274CUB00 | 0% | L5 TM | 38 | 26 September 2004 | LT52330762004254CUB00 | 9% | L5 TM |
| 9 | 25 March 1989 | LT52330761989084CUB00 | 35% | L5 TM | 39 | 05 March 2005 | LT52330762005064COA00 | 7% | L5 TM |
| 10 | 17 September 1989 | LT52330761989260CUB00 | 18% | L5 TM | 40 | 29 September 2005 | LT52330762005272CUB00 | 29% | L5 TM |
| 11 | 28 March 1990 | LT52330761990087CUB00 | 23% | L5 TM | 41 | 08 March 2006 | LT52330762006067CUB02 | 0% | L5 TM |
| 12 | 20 September 1990 | LT52330761990263CUB00 | 20% | L5 TM | 42 | 16 September 2006 | LT52330762006259COA00 | 20% | L5 TM |
| 13 | 15 March 1991 | LT52330761991074CUB00 | 18% | L5 TM | 43 | 11 March 2007 | LT52330762007070CUB00 | 5% | L5 TM |
| 14 | 23 September 1991 | LT52330761991266CUB00 | 18% | L5 TM | 44 | 19 September 2007 | LT52330762007262CUB00 | 19% | L5 TM |
| 15 | 01 March 1992 | LT52330761992061CUB00 | 0% | L5 TM | 45 | 13 March 2008 | LT52330762008073CUB00 | 10% | L5 TM |
| 16 | 09 September 1992 | LT52330761992253CUB00 | 8% | L5 TM | 46 | 21 September 2008 | LT52330762008265CUB00 | 15% | L5 TM |
| 17 | 20 March 1993 | LT52330761993079CUB00 | 29% | L5 TM | 47 | 16 March 2009 | LT52330762009075COA02 | 1% | L5 TM |
| 18 | 28 September 1993 | LT52330761993271CUB00 | 30% | L5 TM | 48 | 24 September 2009 | LT52330762009267COA02 | 14% | L5 TM |
| 19 | 03 February 1994 | LT52330761994034CUB00 | 28% | L5 TM | 49 | 19 March 2010 | LT52330762010078CUB00 | 6% | L5 TM |
| 20 | 17 October 1994 | LT52330761994290CUB00 | 19% | L5 TM | 50 | 11 September 2010 | LT52330762010254CUB00 | 18% | L5 TM |
| 21 | 26 March 1995 | LT52330761995085CUB00 | 16% | L5 TM | 51 | 22 March 2011 | LT52330762011081CUB00 | 7% | L5 TM |
| 22 | 18 September 1995 | LT52330761995261CUB00 | 26% | L5 TM | 52 | 30 September 2011 | LT52330762011273CUB00 | 11% | L5 TM |
| 23 | 28 March 1996 | LT52330761996088CUB00 | 19% | L5 TM | 53 | 12 April 2013 | LC82330762013102LGN01 | 8% | L8 OLI |
| 24 | 20 September 1996 | LT52330761996264CUB02 | 20% | L5 TM | 54 | 19 September 2013 | LC82330762013262LGN00 | 5% | L8 OLI |
| 25 | 11 February 1997 | LT52330761997042CUB00 | 30% | L5 TM | 55 | 14 March 2014 | LC82330762014073LGN00 | 2% | L8 OLI |
| 26 | 07 September 1997 | LT52330761997250CUB01 | 19% | L5 TM | 56 | 22 September 2014 | LC82330762014265LGN00 | 3% | L8 OLI |
| 27 | 03 April 1998 | LT52330761998093COA00 | 19% | L5 TM | 57 | 17 March 2015 | LC82330762015076LGN00 | 3% | L8 OLI |
| 28 | 10 September 1998 | LT52330761998253COA02 | 18% | L5 TM | 58 | 25 September 2015 | LC82330762015268LGN00 | 3% | L8 OLI |
| 29 | 06 April 1999 | LT52330761999096CUB00 | 0% | L5 TM | 59 | 19 March 2016 | LC82330762016079LGN00 | 2% | L8 OLI |
| 30 | 29 September 1999 | LT52330761999272COA02 | 9% | L5 TM | 60 | 11 September 2016 | LC82330762016255LGN00 | 2% | L8 OLI |

*2.3. Predictive Modelling of Vegetated Surfaces*

2.3.1. Meteorological Information and Water Table Depths

The precipitation and temperature data for the salt meadows of Tilopozo were obtained from historical records (1975–2016) for the Piene meteorological station of the DGA, located 12 km from the study area (23°41′03″S, 68°03′29″W, 2460 m above the mean sea level). Any missing data were calculated using an estimate of the average of the corresponding months [38]. The potential evapotranspiration was calculated using the minimum, maximum and average monthly temperatures from 1975–2016, as well as the monthly solar radiation for the same period, following the proposals by Hargreaves et al. [39] and Samani [40] for hyper-arid zones. Subsequently, the dry and wet seasons were determined by identifying the trimesters with least and most precipitation, respectively [38].

The information on the groundwater level in the salt meadows of Tilopozo was obtained from the water monitoring network of the Early Warning Plan of the MNT-aquifer [9], which has 30 piezometers and observation wells installed in the surroundings of the salt meadows of Tilopozo to control or prevent the negative effects that could result from over-exercising rights to use water from the aquifer which has been granted for the development of industrial activity. Information on cutting-edge piezometers and observation wells are available from 2000, when the monitoring scheme for the salt meadows of Tilopozo began [9].

2.3.2. Construction of Models to Predict Surface Vegetation Cover

The first step in constructing the models consisted of performing covariance and correlation tests on average precipitation, average evapotranspiration and water table depths from piezometers. Data from the observation wells of the water monitoring network were used, along with vegetation cover data for the salt meadows, to determine the best covariance and correlation values and to define statistically significant relationships between the variables [41]. The procedure was performed separately for the annual model, and the dry (September to November) and wet (January to March) season models (Figure 4).

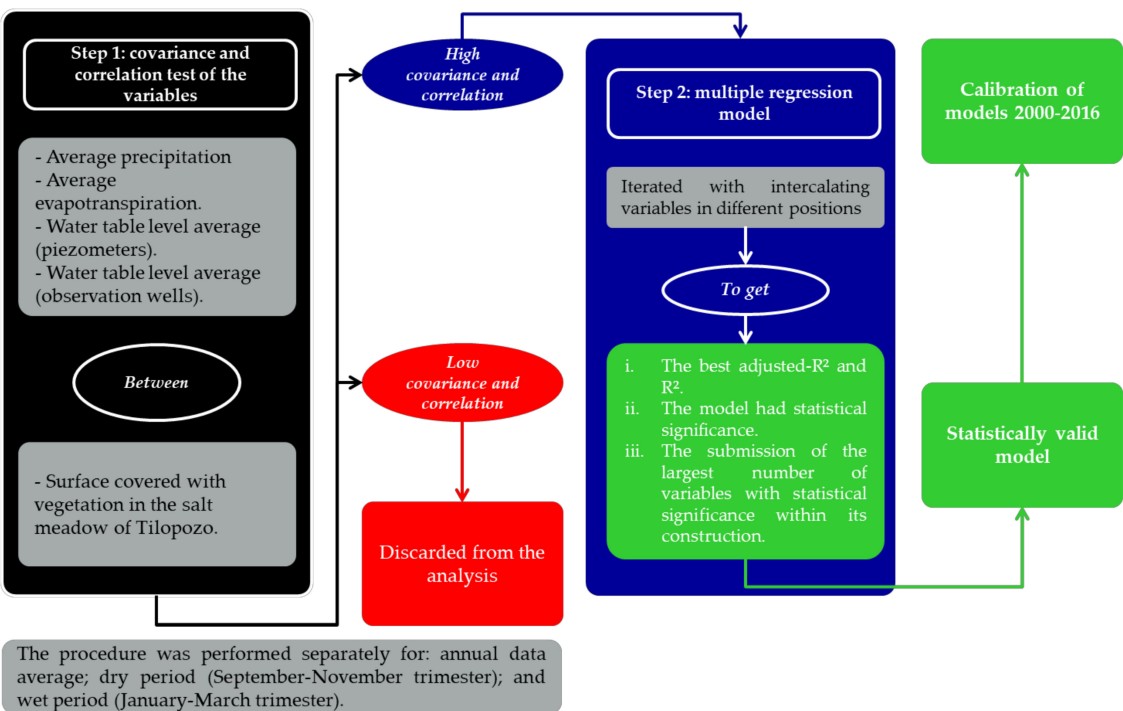

**Figure 4.** Two-step procedure for the selection of valid variables for the construction of multiple regression models, such as predictors of surface covered with vegetation in the salt meadows of Tilopozo, for annual, dry season and wet season data.

The second step was to select and process the variables that showed a high covariance and correlation with vegetation cover using a multiple regression model. The models were iterated by intercalating variables in different positions, after which the best-fit models were selected based on the following criteria:

(i)  models that had the best $R^2$ and $R^2$-adjusted values;
(ii)  models that had high statistical significance; and
(iii)  models that had the greatest number of statistically significant variables.

Statistical significance was evaluated using the Fisher's F-test, with a confidence interval of 95% (Figure 4). Three prediction models of vegetation cover were constructed:

(i)  an annual prediction model;
(ii)  a prediction model for the dry period; and
(iii)  a prediction model for the wet period.

Models were calibrated for the period 2000–2016. The models were run using rainfall, evapotranspiration and water table depth data. Variations between the estimated (from satellite images) and modeled surface vegetation cover were observed.

Models predicting vegetation cover for the salt meadows were validated by comparing the area estimates from satellite images with those obtained from the models. With that information a trend line was adjusted, and the coefficient of determination was obtained.

## 3. Results

### 3.1. Vegetation Cover in the Tilopozo Salt Meadows between 1985 and 2016

The average decrease in water table depths in the well and piezometers were 8 cm between 1985 and 2016. The average annual rate of decrease for all wells (*TP*) and piezometers (*TPZ*) was −0.013 cm year, ranging from −0.8 cm year at piezometer $TPZ_2$ to −0.0004 cm year at piezometer $TPZ_{10}$ (Table 2). The surface vegetation cover, determined from satellite images, in the salt meadows of Tilopozo showed oscillating behavior, as a function of time. On average, 100.43 ha was covered with vegetation, with a maximum area of 166.05 ha in March 1995, during the wet season, and a minimum of 60.48 ha in September, during the dry season of 2009 (Figure 5). A 34 ha decrease in the satellite image-derived estimates of the vegetated area of the Tilopozo salt meadows was recorded between 1985 and 2016. The differences in the greatest and lowest areal extents of vegetation were higher for the dry periods (44 ha) than for the wet periods (24 ha) (Figure 5). The rates of vegetation loss differed markedly before and after 2002. From 1985 to 2002 there was a net increase of 15 ha, equating to 0.88 ha year. During this time, the maximum area of vegetation was 166 ha in March 1995, and the lowest area was 62 ha in September 1992 (Figure 6a). Meanwhile, from 2003 to 2016, the decrease in the vegetated area was lower (−0.54 ha/year); the net decrease was 6 ha (Figure 6b).

**Table 2.** Average annual water table depths in piezometers (*TPZ*) and observation wells (*TP*) used for the construction of a model for the prediction of the surface covered with vegetation in the salt meadows of Tilopozo, during the period, 2000–2016.

| Year | Phreatic Level of Observation Points (Meters above Sea Level) (Piezometers = *TPZ*; Wells = *TP*) | | | | | | | |
|---|---|---|---|---|---|---|---|---|
| | $TPZ_{2A}$ | $TPZ_{2B}$ | $TPZ_{2C}$ | $TPZ_3$ | $TPZ_4$ | $TPZ_8$ | $TPZ_{12}$ | $TP_2$ |
| 2000 | −1.53 | −1.08 | −0.99 | −1.21 | −0.96 | −0.65 | −0.84 | −3.78 |
| 2001 | −1.44 | −1.04 | −0.91 | −1.15 | −0.88 | −0.61 | −0.8 | −3.79 |
| 2002 | −1.52 | −1.13 | −0.99 | −1.23 | −0.94 | −0.66 | −0.85 | −3.80 |
| 2004 | −1.53 | −1.17 | −1.02 | −1.22 | −0.91 | −0.66 | −0.85 | −3.84 |
| 2005 | −1.51 | −1.16 | −1.00 | −1.21 | −0.93 | −0.64 | −0.82 | −3.84 |
| 2006 | −1.58 | −1.19 | −1.04 | −1.24 | −0.99 | −0.65 | −0.85 | −3.86 |

**Table 2.** *Cont*.

| Year | Phreatic Level of Observation Points (Meters above Sea Level) (Piezometers = *TPZ*; Wells = *TP*) | | | | | | | |
|---|---|---|---|---|---|---|---|---|
| | $TPZ_{2A}$ | $TPZ_{2B}$ | $TPZ_{2C}$ | $TPZ_3$ | $TPZ_4$ | $TPZ_8$ | $TPZ_{12}$ | $TP_2$ |
| 2007 | −1.58 | −1.20 | −1.07 | −1.26 | −0.98 | −0.67 | −0.86 | −3.9 |
| 2008 | −1.55 | −1.16 | −1.00 | −1.23 | −0.95 | −0.64 | −0.84 | −3.93 |
| 2009 | −1.54 | −1.18 | −1.03 | −1.21 | −0.96 | −0.63 | −0.84 | −3.90 |
| 2010 | −1.57 | −1.20 | −1.06 | −1.25 | −1.04 | −0.66 | −0.87 | −3.92 |
| 2011 | −1.58 | −1.21 | −1.08 | −1.25 | −1.05 | −0.67 | −0.88 | −3.92 |
| 2013 | −1.55 | −1.18 | −1.05 | −1.21 | −0.97 | −0.64 | −0.85 | −3.94 |
| 2014 | −1.55 | −1.24 | −1.08 | −1.24 | −1.00 | −0.68 | −0.89 | −3.96 |
| 2015 | −1.52 | −1.19 | −1.05 | −1.21 | −0.92 | −0.68 | −0.87 | −3.96 |
| 2016 | −1.61 | −1.26 | −1.12 | −1.28 | −0.97 | −0.71 | −0.93 | −4.00 |

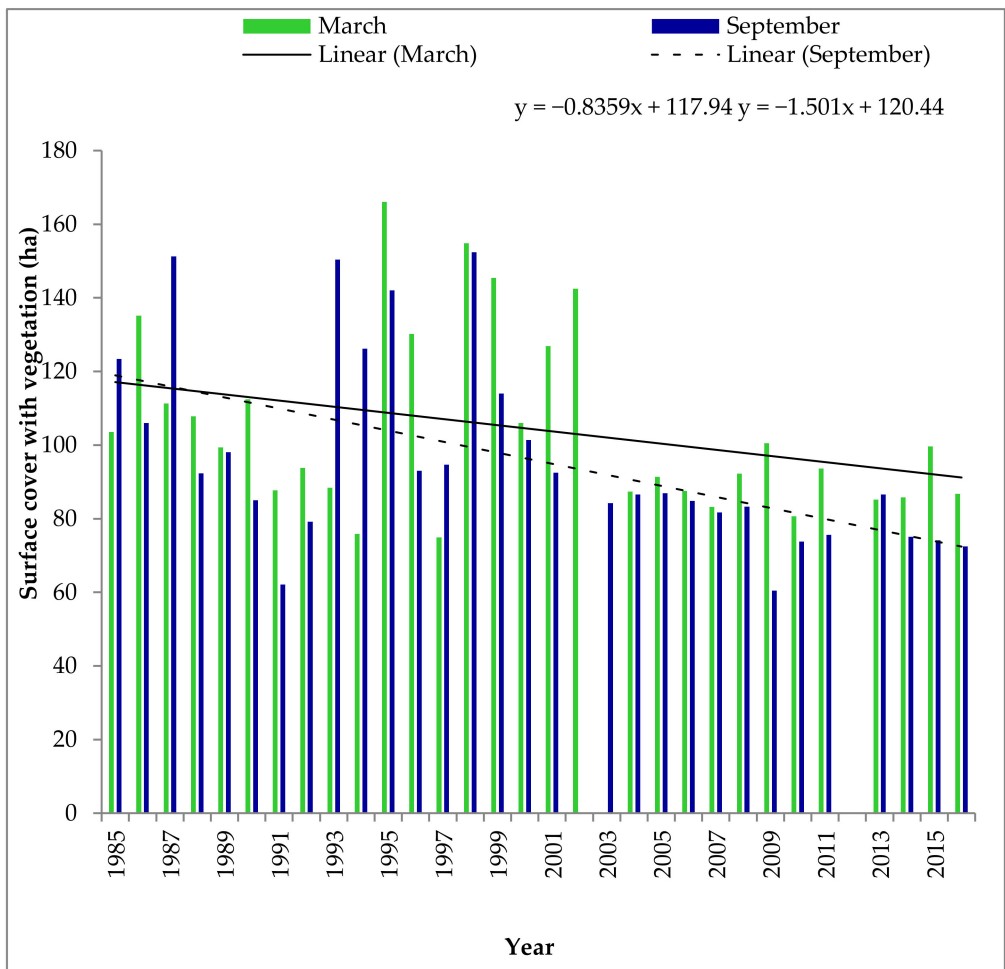

**Figure 5.** Evolution of the area of vegetation cover of the Tilopozo wetlands (from Landsat TM and OLI-derived normalized differential vegetation index (NDVI) data) for March and September between 1985 and 2016.

### 3.2. Prediction Models of the Vegetation-Covered Surface

The prediction model for the vegetation-covered surface used water table depth records from eight piezometers and observation wells (Table 2). This is less than the total number of piezometers and observation wells in the monitoring network, because some of them were rejected because of known errors [30].

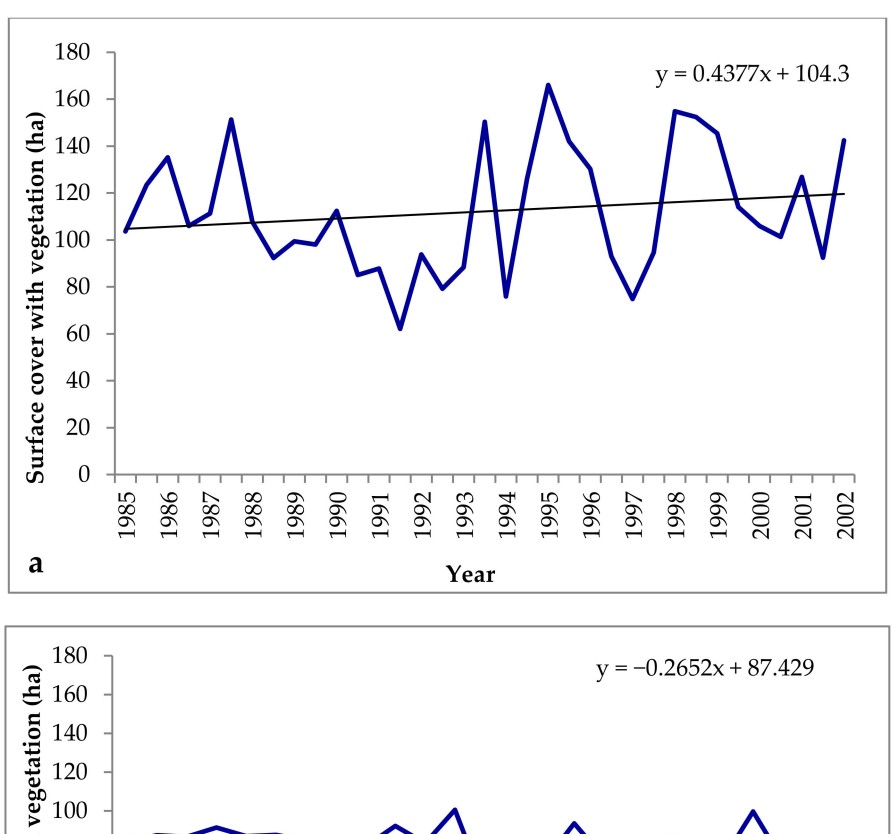

**Figure 6.** Trends in vegetated areas of the Tilopozo salt meadows 1985–2002 (**a**) and 2003–2016 (**b**).

### 3.2.1. Prediction Model of Wet Period Vegetation Cover

The prediction model of the land surface covered with vegetation for the wet period, from January to March, was formed using the trimester averages of precipitation, evapotranspiration and the water table levels from piezometers ($TPZ_{2A}$, $TPZ_{2B}$, $TPZ_3$, $TPZ_4$, $TPZ_8$ and $TPZ_{12}$), and the observation well $TP_2$. Ninety-seven iterations were performed to determine the best fit to Equation (1).

$$Co_w = 456 + 1.35\,Pp - 0.816\,ETp + 266\,TPZ_{2A} + 103\,TPZ_{2B} - 687\,TPZ_3$$
$$+176\,TPZ_4 + 1160\,TPZ_8 - 970\,TPZ_{12} + 113\,TP_2, \tag{1}$$

where $Co_w$ is the surface covered with vegetation in the wet period (ha); $Pp$ is the precipitation for the wet period (mm); $ETp$ is the potential evapotranspiration for the wet period (mm); $TPZ_{2A}$, $TPZ_{2B}$, $TPZ_3$, $TPZ_4$, $TPZ_8$ and $TPZ_{12}$ are the water table levels for the wet period (m); and $TP_2$ is the water table depth for observation well for the wet period (m).

In this model, six (of 10) variables showed statistical significance (the constant, $Pp$, $TPZ_3$, $TPZ_4$, $TPZ_8$, and $TPZ_{12}$). The model, in its entirety, was statistically significant ($p < 0.05$), in addition to showing strong correlations for the estimated values with vegetated areas mapped from satellite images and modeled data ($R^2 = 97.10\%$ and $R^2$ (adj.) = 91.90%).

### 3.2.2. Prediction Model of Dry Period Vegetation Cover

The equivalent prediction model for the dry period of July–September used trimester averages of precipitation, evapotranspiration and the water table levels from piezometers, $TPZ_{2B}$, $TPZ_4$. $TPZ_8$, and $TPZ_{12}$. Ninety-six iterations were performed to determine the best fit to Equation (2).

$$Co_d = 38 + 6.23\ Pp + 1.69\ ETp + 277\ TPZ_{2B} - 64.30\ TPZ_4 - 464\ TPZ_8 - 970\ TPZ_{12} \tag{1}$$

where $Co_d$ is the surface covered with vegetation in the dry period (ha); $Pp$ is the precipitation for the dry period (mm); $ETp$ is the potential evapotranspiration for the dry period (mm); and $TPZ_{2B}$, $TPZ_4$, $TPZ_8$ and $TPZ_{12}$ are the water table levels for the dry period (m). In this model, three (out of seven) variables showed statistical significance ($Pp$ and to piezometers readings from $TPZ_{2B}$). The model, in its entirety, was statistically significant ($p < 0.05$), in addition to showing strong correlations for the estimates of vegetated areas mapped from satellite images and the modeled data ($R^2$ = 81.60% and $R^2$ (adj.) = 67.90%).

### 3.2.3. Prediction Model of Annual Vegetation Cover

The annual prediction model of the surface covered with vegetation used average annual precipitation and evapotranspiration, and depths to the water table from the $TPZ_{2C}$ piezometer and the observation well $TP_2$. Ninety-nine iterations were performed to determine the best fit to Equation (3).

$$Co_y = -683 + 5.25\ Pp + 0.33\ ETp - 63.60\ TPZ_{2c} + 184\ TP_2, \tag{3}$$

where $Co_y$ is the surface covered with vegetation for the annual period (ha); $Pp$ is the annual precipitation for the period (mm); $ETp$ is the potential evapotranspiration for the annual period (mm); $TPZ_{2c}$ is the piezometers for the annual period (m); and the level for the observation well $TP_2$ for the wet period (m). In this model, three (out of five) variables showed statistical significance (the constant, $Pp$, and $TP_2$). The model, in its entirety, was statistically significant ($p < 0.05$). The correlations between the estimated values for vegetated area mapped from the satellite images and the modeled data were high ($R^2$ = 82.20% and $R^2$ (adj.) = 75.10%).

Differences between estimated vegetated area from the annual model and the values obtained from satellite images were minimal, and were less variable than the dry and wet period models. Extreme precipitation events, such as those registered in 2002 and 2015, produced differences in the estimates of the vegetated area mapped from satellite images and modeled values. However, the trends shown in Figure 7 were similar.

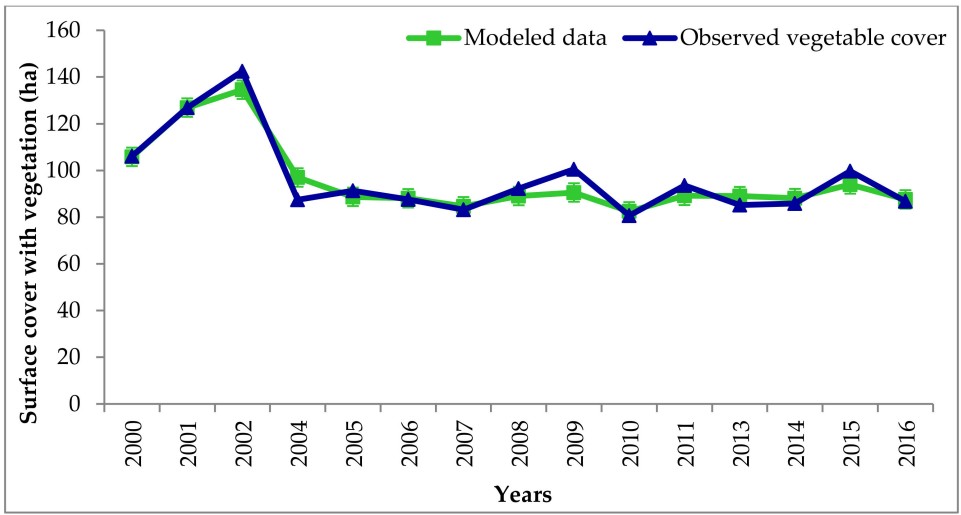

**Figure 7.** Modeled vegetation areas (annual model) and those estimated from satellite images for the Tilopozo salt meadows, 2000–2016.

### 3.2.4. Model Validation

The estimates of vegetated areas from satellite imagery were compared to outputs from the predictive models Equations (1)–(3). For the wet period, the average difference between the vegetated area estimates and the modeled data was 0.91 ± 4.91 ha. The greatest differences were observed in 2004 and 2009, where strong oscillations in the surface covered with vegetation were observed.

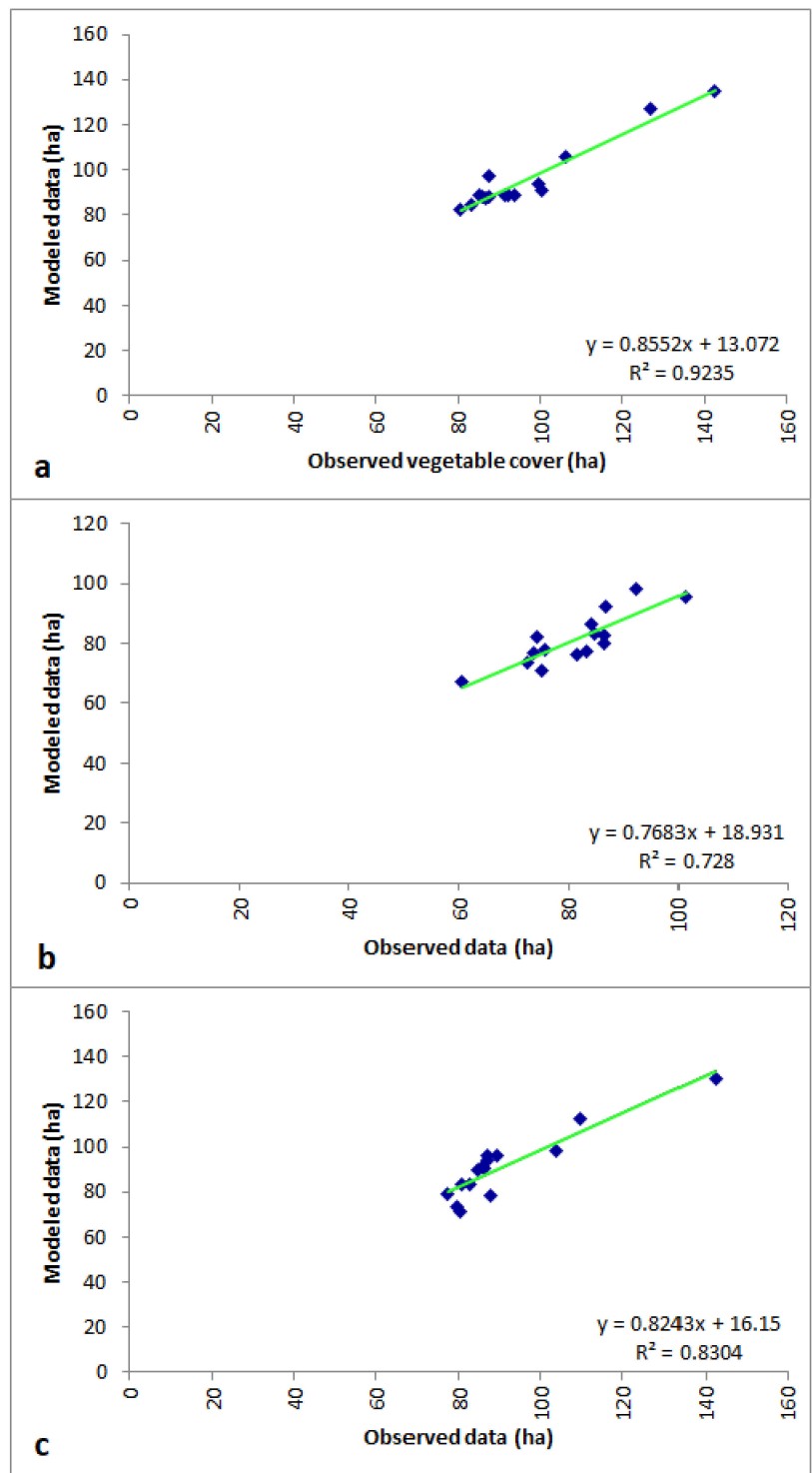

**Figure 8.** Model validations based on modeled and observed data for the wet period model (**a**), dry period model (**b**), and the annual model (**c**).

The correlation between the vegetation area estimates and the modeled data was $R^2 = 92.40\%$ (Figure 8a). The model for the dry period had an average difference of $0.09 \pm 5.11$ ha between the vegetated areas estimates from satellite images and modeled data, although the differences in each year were greater than in the wet period model. The correlation coefficient between the two estimates was $R^2 = 72.80\%$ (Figure 8b). The average difference between vegetated areas mapped from satellite images and the modeled data was $0.18 \pm 6.89$ ha. The major differences between the estimated and modeled data occurred when extreme variations in the surface covered with vegetation were observed. These were the result of extreme climate events, and show a correlation of $R^2 = 83.04\%$.

## 4. Discussion

The differences observed in the changes in water table depths can be explained by the structure of the aquifer and the soils of the salt meadows, which were described as having different permeabilities, which facilitates or hinders the movement of water [42]. Furthermore, Vyacheslav et al. [43] observed that the well and the piezometers upstream of the salt meadows of Tilopozo presented greater water table depths due to the cone of depression produced by the extraction of waters in the upper part of the MNT-aquifer.

After ten years of pumping in the upper part of the MNT-aquifer, there was a decrease in the water table in the Tilopozo's salt meadows [44,45]. This in turn generated a change in the observed vegetal coverage rates.

Thes differences in decreases in the water table (due to variations as the permeability and fissures in the soil) imply an increase in the aeration zone and a decrease in the saturated zone. This affects the soil moisture availability of the highest points in the salt meadows, limiting the soil moisture available for the transitional hygrophytes. The specific responses of these species will depend on adaptive strategies and root systems [46–48]. Consequently, lowering of the water table is reflected in the loss of surface areas covered with vegetation.

The surface covered with vegetation in the salt meadows of Tilopozo between 1985 and 2002 showed both intra- and inter-annual oscillations (Figure 5a). This behavior could be due to the fact that the analysis of the vegetation using NDVI can be influenced by the phenological stage of the species present in the study area, which may modify the estimation of the surface covered with vegetation [20,26]. Concerning the species in the salt meadows of Tilopozo, little information is available on their spectral behavior according to their phenological stage. One of the most important and extensive plants in the salt meadows of Tilopozo is *S. americanus*. However, Langley and Megonigal [49] indicated that determining the phenological stage of *S. americanus* using NDVI is very complex, because the biomass is brown, and this generates very low NDVI values, which can be interpreted as bare soil. In addition, as *S. americanus* plants increase in height, their leaf areas decrease, which impacts NDVI in a counterintuitive manner, because higher biomass leads to lower NDVI values [50].

NDVI has been used widely to estimate vegetation biomass, but its use has limitations in arid lands [51]. For example, vegetation is not always green, and therefore, it is difficult to detect using NDVI [51,52]. Vegetation can also be confused with soil [53], since darker soils can have higher NDVI than vegetation [54]. Qu et al. [19] developed a multiple regression model that determined the response of vegetation to climatic factors, and complex relationships that indicated lags between the occurrence of a meteorological event, and the observable effects on the NDVI in areas with greater precipitation were obtained. Meanwhile, du Plessis [16] pointed out that statistical relationships between vegetation cover and precipitation in arid-zone grasslands are deficient because there are factors that determine vegetation behavior that cannot be measured from satellite imagery such as slope, soil moisture and water table depth. Sorrell et al. [55] also identified water table depth as a factor that determines hygrophyte species cover.

Wetlands with small areal extents have been appropriately delimited through the use of NDVI [16,17,36,37,56]. These studies have been confirmed by Dong et al. [18], who developed a wetland map-based in NDVI and obtained accuracies between 83–87%.

In arid-zone wetlands in Spain, Domíngez-Biesiegel et al. [35] used an NDVI value of 0.20 for the determination of vegetation. Moreover, Pan et al. [21] evaluated Chinese wetlands using an NDVI threshold of 0.13, the same value used in this investigation.

Other authors working in similar situations highlighted that the NDVI offers the best estimates of percentage plant cover and can be used in a wide range of vegetation densities on arid and semi-arid zone meadows [19,56,57]. Purevdorj et al. [57] highlighted that SAVI is a better indicator of vegetation cover at low densities than NDVI. However, it is necessary to have soil correction factors to calculate SAVI, which restricts its application [56,58].

The vegetation in the salt meadows is likely to reduce the NDVI values of Tilopozo, and can be impacted by various activities, such as the extraction of water in the upper part of the aquifer [9], cultivation and grazing [24], and even the uncontrolled burning of the vegetation present in the salt meadows. Burning is not formally recorded, but was observed by the authors during field campaigns (Figure 9).

The impact of climatic variables on vegetation cover in the salt meadows of Tilopozo was less than the human actions listed above, and this may be due to the fact that it is hygrophilous, azonal vegetation, which is not dependent on climatic conditions, but rather depends primarily on the availability of water coming from aquifers, or other hydrological characteristics of the study area, such as streamflow in rainy periods [7,59]. A high dependence of vegetation on water table depths was observed in all prediction models; see Equations (1)–(3). This inference is supported by high correlation coefficients between vegetation cover and water table depths recorded in the water monitoring network.

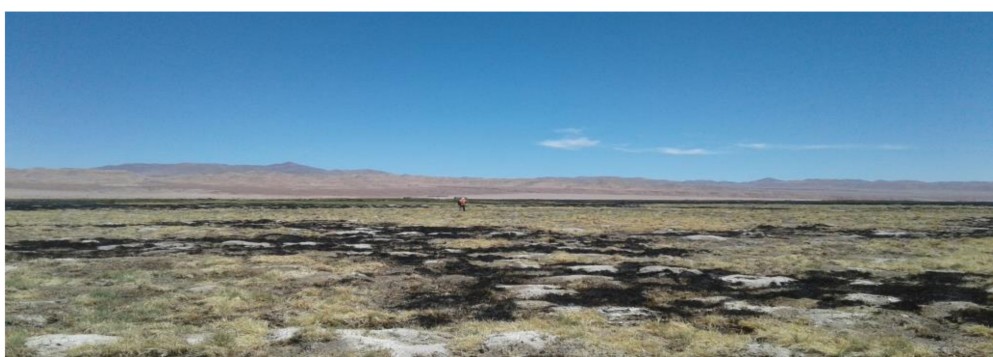

**Figure 9.** *Schoenoplectus americanus* meadows at Tliopozo that were damaged by an intentional fire in November 2017; authors' photograph.

These characteristics are valid for multiple regression models with different objectives [13,60,61]. The statistical performance of the models developed in this research for the wet, dry and annual periods was corroborated by obtaining the best $R^2$ and $R^2$-adjusted values and their statistical significance. Consequently, the predictive models were well balanced and very close to the estimated values (Figure 6), with the average differences between modeled and satellite image-derived vegetation area estimates not exceeding 1 ha in any of the 90 surfaces investigated for the salt meadows of Tilopozo. The behavior of the prediction models was affected by the occurrence of abrupt increases and decreases ing vegetation cover. These variations are probably due to extreme weather events, such as the 40.5 mm of precipitation recorded in 24 h in 2002, which can be compared to the mean annual precipitation of 20 mm. Extreme weather phenomena may activate growth in many of the species present in the area [62].

The occurrence of such events affects the efficiency of the models, and can lead to under- and overestimation in predicted values [63]. The accuracy and the use of data from the water monitoring network prediction models of vegetation cover can be used in follow-up work in wetlands like those at Tilopozo. The follow-up work on vegetation behavior, through the use of a water monitoring network data, should enable the determination of the status of a wetland, and be able to integrate human pressures [64].

The existing research into the wetlands studied has focused on evaluating their composition and structure [24,65,66], but there has been little research into the background status of these wetlands against which predicted future states can be compared [63]. According to Liu et al. [64], in the Ordos Larus Relictus National Nature Reserve, China, there was a change in the structure of the wetland between 1990 and 2014 due to climate change and the indiscriminate extraction of water. The free water column decreased by 83.7%, which resulted in vegetation fluctuations. Liu et al. [64] also predicted that, by 2100, the free water column will have almost disappeared, and that vegetation present will be reduced by 22.3%.

## 5. Conclusions

The historical analysis of the surface covered with vegetation in the Tilopozo salt meadows showed a loss of 34 ha between 1985 and 2016, which equates to an average annual loss of 1.08 ha year$^{-1}$. These changes were mainly correlated with water table variations. A possible lag of ten years between the start of the pumping in the upper part of MNT-aquifer and the first observable effects on the vegetation cover in the Tilopozo salt meadows was also noted.

The behaviour of the vegetation cover was less dependent on meteorological conditions than on groundwater levels. Nonetheless, the existence of extreme precipitation events generated significant increases in vegetation coverage, which were mainly induced by the activation of germplasm or the activation of transitional hygrophilic species in the salt meadow ecotone.

The three predictive models showed high correlations between the modeled vegetation cover and vegetation area estimates from satellite data. The models comply with the objective of being simple to build and based on easily accessible information.

The annual model had high correlation ($R^2$ = 83.04%), and is proposed as a tool that will allow the behavior of the vegetation covered surface estimated, using only meteorological information and the groundwater levels in the piezometers and and the observation well.

The development of the prediction models has facilitated an understanding of some of the reasons for the fluctuations on the vegetation cover of the Tilopozo salt meadows, in particular climate variability and human actions. Climate change may ultimately be very important as it affects water availability in the high Andes [67]. Prediction models like the ones developed in this research will allow the status of salt meadows and other types of wetlands in hyper-arid areas to be predicted. They can be useful tools in conservation planning and management, as well as in developing a deeper understanding of the behavior of vegetation. The models developed in this research will, in particular, help improve water management in the MNT-aquifer and help lessen the impact of pumping on the Tilopozo wetland.

**Author Contributions:** J.S., C.R.-F., and M.P. conceived and designed this study; J.S. and C.R.-F. executed the methodology; J.S., C.R.-F. and M.P. wrote the paper.

**Conflicts of Interest:** The authors declare no conflicts of interest.

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
