# Peer review of "A Model for Estimating the Vegetation Cover in the High-Altitude Wetlands of the Andes (HAWA)"

_land, doi:10.3390/land8010020_

Round 1

Reviewer 1 Report

Thanks for the opportunity to review this manuscript.  The study estimated the salt meadow covered surface was using normalized difference vegetation index (NDVI) for the period from 1985 to 2016. Then, models were built to predict the covered surface with vegetation using local meteorological data and an underground water monitoring network. A decrease of 37 ha in the surface covered with vegetation in a period of 32 years was determined for the salt meadow of Tilopozo. The models for the prediction of the surface covered with vegetation for the dry and wet annual period showed successful results with correlations between 72.80% and 92.40%, between the modelled and the observed data, respectively. Authors further concluded that the generated models can be useful tools for the study and conservation of the salt meadow of  Tilopozo by providing relevant information on the state of the vegetation and allowing the projection of future behaviour as a function of climate information, and a water monitoring network. However, although the idea is very brilliant, the performance of NDVI should be compared with other indices. The area under study seems to have a lot of bare surfaces hence, background reflectance is likely to a be a challenge. Maybe authors should rather select images for a particular date and apply different indices and see how the results compare to those of NDVI and then proceed with the use of NDVI.

Secondly, there is a need for in-depth interrogation of literature to strengthen the introduction and the discussion. So far the two sections are generally weak and lack details

The use of abbreviations should be consistent throughout the manuscript. In some instances, authors use HAW and HAWA interchangeably and this is confusing.

There is also a need to improve on grammar and spelling mistakes. 

Author Response

Answer Comments land-381043

We thank the reviewer for their valuable comments. These comments are very constructive, and will help us to improve the manuscript, specifically in terms of clarifying our methodology and the goal of this paper. We address the reviewer’s concerns in this letter, and corresponding changes will be made to improve the manuscript.

Reviewer 1.

1.             The performance of NDVI should be compared with other indices.

A bibliographic analysis was carried out, finding that similar experiences have been used by the NDVI for their analyzes, given the particularities of the study area: dense and agglomerated vegetation in limited sectors (not scattered). This information has been added in the introduction, method and discussion.

In addition, to reinforce the point, we have made a small analysis. The summary of the results you can see in the response to the third comment of the reviewer. All details can be seen in: "Comparison of vegetation cover indexes for Tilopozo".

2.             The area under study seems to have a lot of bare surfaces hence, background reflectance is likely to a be a challenge.

Yes, effectively, to estimate surface covered with vegetation was a challenge, for this reason a NDVI of 0.13 was employed. This value was obtained from Chuvieco (2008). Besides, the group experience in the study area and in the arid and semi-arid land in the North of Chile (Montalva et al., 2015) allows selecting this value for vegetation determination. Castro et al. (2014) utilized a similar value in the Atacama Desert (Copiapo) for vegetation determination. They determined this value as threshold discarding the zones without vegetation according its presence or absent. In this study, the zones without vegetation were discarded compating results from NDVI with real color images and the knowledge about the study area.

3.             Maybe authors should rather select images for a particular date and apply different indices and see how the results compare to those of NDVI and then proceed with the use of NDVI.

A comparative analysis was carried out among 8 indices for the determination of vegetation cover in the Tilopozo area. The results obtained indicate that due to the concentration of the vegetation at three specific points (outcrops of fresh water), the variation between indices is low. The above is particularly notorius when we comparing the CTVI, SAVI and TTVI indices, respect to NDVI; coinciding in the identification of the surface with and without vegetation, in 100%, 99.05% and 98.40%, respectively.

Therefore, in the case of the Tilopozo study, using the NDVI does not imply a under-representation or overrepresentation of the surface covered with vegetation. Consequently, we believe that the use of the NDVI for the study is appropriate, as has been done by many other authors for similar studies (added in the paper corrections).

The procedural details and the associated literature can be reviewed in the attached report: "Comparison of vegetation cover indexes for Tilopozo".

4.             There is a need for in-depth interrogation of literature to strengthen the introduction.

An in-depth analysis was made to the state of the art respect to the use of GIS instruments and statistical tools, the use of these combinations on the delimitation of wetlands and the application of studies of change of coverage in wetlands and projection tools for conservation

5.             There is a need for in-depth interrogation of literature to strengthen the discussion.

The discussion was enriched, adding information regarding the decreases of the water table, the variation of surfaces, the use of the NDVI as a tool to delimit the surface of wetlands, results of other similar investigations regarding the use of regression models and obtaining indicators statistics such as R2 and R2-adjusted.

6.             The use of abbreviations should be consistent throughout the manuscript. In some instances, authors use HAW and HAWA interchangeably and this is confusing.

Answered. It was unified in HAWA.

7.             There is also a need to improve on grammar and spelling mistakes.

We will review

Reviewer 2 Report

Revision of the manuscript ID: land-381043 “A Model for Estimation of Vegetation Cover at High Altitude Wetlands Andean (HAWA)”.

Comments to the authors

General comments

The article deals with the use of NDVI index to quantify the vegetated area of Tipoloco salt meadow and comparison with a regression model obtained from groundwater data and meteorological data. The idea of the article is original and merits publication. However, the article is not ready for publications, it needs to be improved in terms of writing; and authors should order the given information in the different sections. For example, the study area characteristics appear sparse and uncomplete in different sections.

The models are not mentioned: are they regression relationships? The observed data are not mentioned. NDVI are estimated data, not observed. The NDVI and vegetation relationships must be explained, their consistency, and support to be considered as “ground data”.

The abstract do not reflects the paper content. The problem and objective are not clearly stated. Is your objective the monitoring of vegetation area? Which type of vegetation?

There is not a revision of the state of the art in terms of remote sensing applied to the wetlands vegetation monitoring. Please, rewrite first paragraph of Introcuction. Is the beginning and it must be clear and more forceful.

Specific comments

L43. Clarify what is the line of vegetation growth. Is it altitude?

L44. The limit of …snow: the upper or the lower limit? Clarify what does it mean. Height plains: Do you refer to high plains?

L4. Indicate the salt meadow you refer to.  It is plural or singular? Please revise.

L 47. Indicate the groundwater depth, max and min.

L49. This sentence should be the beginning, what and where.

L50 What depends? The surface water? The ecosystem in general ?Too vague.

L58. What “the hydrological processes “ means? It seems the sentence is not complete.

L64-65 What does it mean: “impact about the vegetation”

L66-67 what type of vegetation is studied by the author mentioned?

L68 Rewrite please. The sentence is not understandable. Here you stated the problem to be mentioned in abstract

L71 I do not understand “The temporal behaviour of the cover with the vegetation”

Which type of vegetation are you studying? Is seasonal vegetation or perennial?

Is the NDVI threshold used in other vegetation types useful here? This must be explained, together with the spectral and vegetative difference between dry and wet season. Why do you use the average of the two vegetation maps if you apply the model separately for the two seasons? A map with the vegetation extent (average and variability) in the two seasons and the location of wells/piezometer would be very useful to understand the article.

L81. Please, replace “Temporal behaviour” with other more appropriate terms. Do you refer to the seasonal (or annual or decadal) changes of vegetation area extent?

L93. Please refer to groundwater level

Meteorological data, averages, are needed when describing the study area.

Please locate the weather station in Figure 1.

L96. Eliminate “in Spanish”

L102 You should explain “the behaviour of the precipitation” meaning. I recommend eliminate “behaviour” from all the sentences in the manuscript.

L114. Here we understand that this is an objective: to predict vegetation

L149 “a trend… between the years” Please clarify.

Section 2.3.2. should be explained with a schema. It is not clear and it is not easy to follow.

L168. The title of the section does not match the content, which is about the climatic characterization of the study area and (some) parameters used /not used for the regression model.

L169-170 This must be included in the description of the study area.

Fig 4. Please, locate the station in the map of Figure 1. In line 571, please, use “monthly  precipitation-evapotranspiration (mm) for the period….”

Figure 5: please, use “predicted and observed surface area covered by vegetation…..”

L176. Explain anomalous data. It can be eliminated from the study? and then Table 2 is necessary?

Figure 2. There is a high interannual variability of vegetation cover. It must be explained in the results section.

L185. Please, explain “the quarterly average of precipitation”

L190. Is the potential evapotranspiration useful taking into account the low rains in this area? Please, include the method of ETP calculation.

L345. Eliminate “appropriate”.

The article must include a commented and discussed synthesis of NDVI values and areas, and a synthesis of correlations, R2 values and errors obtained.

Author Response

Answer Comments land-381043

We thank the reviewer for their valuable comments. These comments are very constructive, and will help us to improve the manuscript, specifically in terms of clarifying our methodology and the goal of this paper. We address the reviewer’s concerns in this letter, and corresponding changes will be made to improve the manuscript.

Reviewer 2.

1.     The study area characteristics appear sparse and uncomplete in different sections.

Answered. Everything has been compiled in description of the studio area.

2.     The models are not mentioned: are they regression relationships?

Answered. This is explained in section 2.3.2. Construction of the Model of Prediction of the Surface Vegetation Cover

3.     The observed data are not mentioned.

In this study did not use observed data. Information obtained from satellite images was considered as the baseline for surface covered with vegetation.

4.     NDVI are estimated data, not observed.

The observation is appreciated. The change was made in the places that corresponded.

5.     The NDVI and vegetation relationships must be explained, their consistency, and support to be considered as “ground data”.

The information was complemented with other experiences carried out in a hyper-arid context. Also, in the discussion increased the use of NDVI in arid contexts.

6.     The abstract do not reflects the paper content.

The summary was redrafted.

7.     The problem and objective are not clearly stated. Is your objective the monitoring of vegetation area? Which type of vegetation?

It has been redrafted

8.     There is not a revision of the state of the art in terms of remote sensing applied to the wetlands vegetation monitoring.

An in-depth analysis was made to the state of the art respect to the use of GIS instruments and statistical tools, the use of these combinations on the delimitation of wetlands and the application of studies of change of coverage in wetlands and projection tools for conservation

9.     Rewrite first paragraph of Introduction. Is the beginning and it must be clear and more forceful.

Answered. We eliminate the given paragraph that does not contribute in the context of this investigation.

10. L43. Clarify what is the line of vegetation growth. Is it altitude?

Answered. Changed by: "maximum altitude of vegetation growth"

11. L44. The limit of …snow: the upper or the lower limit? Clarify what does it mean. Height plains: Do you refer to high plains?

The limit of …snow: the upper or the lower limit? Clarify what does it mean. 

Answered. Changed by: "under the zero isotherm".

Height plains: Do you refer to high plains? 

Answered. Changed by: "high plains”.

12. L44. Indicate the salt meadow you refer to.  It is plural or singular? Please revise.

In this specific case, as we speak of salty wetlands, it is correct to use plural.

13. L 47. Indicate the groundwater depth, max and min.

The paragraph makes a general reference about ecosystems, so there is no established value that is common to all. However, an example was added between parentheses, about the particular situation of Tilopozo: "(eg the depth of the water table in Tilopozo salt meadow, and depending on the area, it is possible to find it between 0 m and 3.65 m deep, with an average of 1.82 m)".

14. L49. This sentence should be the beginning, what and where.

A modification of the introduction was made.

15. L50 What depends? The surface water? The ecosystem in general? Too vague.

Answered. Changed by: "and the presence of the ecosystem depends on the discharge of".

16. L58. What “the hydrological processes “ means? It seems the sentence is not complete.

Answered. Changed by: "and the changes of wetland ecosystems product of the hydrological variations".

L64-65 What does it mean: “impact about the vegetation”.

Answered. Changed by: "the loss of vegetation due to hydrological variations".

17. L66-67 what type of vegetation is studied by the author mentioned?

Answered. Changed by: "the loss of vegetation in marsh and meadow wetland, due to hydrological variations in the study area".

18. L68 Rewrite please. The sentence is not understandable.

It has been redrafted

19. L71 I do not understand “The temporal behaviour of the cover with the vegetation” Which type of vegetation are you studying? Is seasonal vegetation or perennial? Is the NDVI threshold used in other vegetation types useful here? This must be explained, together with the spectral and vegetative difference between dry and wet season. Why do you use the average of the two vegetation maps if you apply the model separately for the two seasons? A map with the vegetation extent (average and variability) in the two seasons and the location of wells/piezometer would be very useful to understand the article.

L71 I do not understand “The temporal behaviour of the cover with the vegetation” Which type of vegetation are you studying? 

Answered: Changed by: "behavior of the salt meadow in Tilopozo between 1985 and 2016".

Is seasonal vegetation or perennial? Is the NDVI threshold used in other vegetation types useful here? 

These questions are answered in a change of the writing and the description of the study area.

This must be explained, together with the spectral and vegetative difference between dry and wet season. 

Resolved in the information added to the description of the study area.

Why do you use the average of the two vegetation maps if you apply the model separately for the two seasons? A map with the vegetation extent (average and variability) in the two seasons and the location of wells/piezometer would be very useful to understand the article. 

Resolved in new figure.

20. L81. Please, replace “Temporal behaviour” with other more appropriate terms. Do you refer to the seasonal (or annual or decadal) changes of vegetation area extent?

Answered: Changed by: "Behavior of the Salt Meadow of Tilopozo between 1985 and 2016".

21. L93. Please refer to groundwater level.

Answered. Information was added in the results and in the discussion.

22. Meteorological data, averages, are needed when describing the study area. Please locate the weather station in Figure 1.

An explanation of the study area was added. A new figure was generated that includes what was requested.

23. L96. Eliminate “in Spanish”..

The phrase has been deleted.

24. L102 You should explain “the behaviour of the precipitation” meaning. I recommend eliminate “behaviour” from all the sentences in the manuscript.

Answered: It was changed to: “was determined the dry and wet season, identifying the trimester with less and more precipitation, respectively”.

25. L114. Here we understand that this is an objective: to predict vegetation.

It is correct, one objective was to predict vegetation in the Tilopozo salt meadow.

26. L149 “a trend… between the years” Please clarify.

Answered: It was changed to: "A decrease in the surface covered with vegetation in the period 1985-2016 was noted".

27. Section 2.3.2. should be explained with a schema. It is not clear and it is not easy to follow.

Answered: A simple outline was added to make it easier to understand. The way of explaining the procedure according to the scheme was changed. Figures numbers were also updated.

28. L168. The title of the section does not match the content, which is about the climatic characterization of the study area and (some) parameters used /not used for the regression model.

The meteorological description was eliminated and only the content corresponding to the title of the section was left.

29. L169-170 This must be included in the description of the study area.

Answered The information was transferred to the Study Area section.

30. Fig 4. Please, locate the station in the map of Figure 1. In line 571, please, use “monthly precipitation-evapotranspiration (mm) for the period….”

Corrected.

31. Figure 5: please, use “predicted and observed surface area covered by vegetation…”

It was changed to: “Annual model for the prediction of surface cover with vegetation, comparing with surface estimated from satellite images, for the period 2000-2016 in the Tilopozo salt meadow, Antofagasta region.”

32. L176. Explain anomalous data. It can be eliminated from the study?

Answered: It was explained because they are anomalous: “The data from the groundwater monitoring network led to a number of piezometers and observation wells with anomalous data observations. This is due to altitude variations (meters), contrary to the trend of the piezometers of the environment. These have been reported as measurement errors (CEA, 2015), so that were excluded for the construction of the prediction model of the surface covered with vegetation.”

33.  Then Table 2 is necessary?.

We agree. With the changes made previously, table 2 is no longer required, and has been eliminated.  The numbering has been updated.

34. Figure 2. There is a high interannual variability of vegetation cover. It must be explained in the results section.

Answered. A new figure was added and the description of the study area was improved.

35. L185. Please, explain “the quarterly average of precipitation”.

The section in the study area was re-written avoiding the use of "quarterly". In the method when reference was made to “quarterly”, we use the word "trimester" which is widely used in publications.

36. L190. Is the potential evapotranspiration useful taking into account the low rains in this area? Please, include the method of ETP calculation.

The text was modified, replacing it by: “was calculated using the minimum, maximum and average monthly temperatures of the period 1975-2016, as well as the monthly solar radiation for the same period, following the proposal by Hargreaves & Samani (1985) and Samani, (2000) for hyper-arid zones".

37. L345. Eliminate “appropriate”.

The word has been deleted.

38. The article must include a commented and discussed synthesis of NDVI values and areas, and a synthesis of correlations, R2 values and errors obtained.

The discussion regarding the use of the NDVI as a tool for determining areas was enriched. Also, on regression models, the utility of including multiple variables and the adjusted R2 and R2-values obtained in similar experiences.

Round 2

Reviewer 1 Report

I am pleased to confirm that the manuscript has improved in quality. Authors have addressed all my concerns full and am pleased to indicate that the idea sound novel and authors have managed to clearly highlight the knowledge gap. The selection of methods has been informed by a thorough review of the literature. I think once published, the article will attract a wide readership. So far I would recommend that authors improve on grammar and spelling mistakes. Based on these comments am pleased to accept the manuscript.

Author Response

Answer Comments land-381043

We thank the Academic Editor for their valuable comments. We address the Academic Editor concerns in this letter, and corresponding changes were made to improve the manuscript.

Land-381043- A Model for Estimating the Vegetation   Cover in the High Altitude Wetlands of the Andes (HAWA).

Abstract

Line 17

Change   ‘the agriculture and pasturage of’ to ‘cultivation and grazing by’

Answer

The suggested changes were made. The   text was edited with change control.

Lines 23-24

Rewrite   sentence as ‘The vegetated area of the Tilopozo salt meadows* decreased by 37   ha over 32 years.’ *I noted one   reviewers asked if salt meadow should be plural or singular. I believe you   replied it was a series of salt meadows, so therefore it was plural. But   twice in the abstract you use the singular salt meadow. If you do mean to use   plural change to salt meadows, and check the rest of the manuscript for   consistency.

Answer

The suggested changes were made. The   text was edited with change control.

The word "meadows" was   corrected throughout the text. This remained as a plural.

Line 32

Change   ‘(remote sensors)’ to ‘(remotely sensed)’

Answer

The suggested changes were made. The   text was edited with change control.

Section 1

Lines   42-43

Would   this change not be better?  ‘…(DGA,   2001) and the water table is either at the ground   surface or a few meters below the ground surface, ’

Answer

A change in the text was generated, but   different from the one suggested: “and the water table is either at the   ground surface or a few meters below this”.

Line 50

Could   you replace ‘industrial purposes’ with a list of the industries that are   using this water?

Answer

It was published in the text. Added the   suggested listing.

Line   53-55

Change   first and second sentences to ‘Changes in soil cover* in the area have   regressed against factors that have been found to influence these changes   (Gaitanis, et al. 2015); for example precipitation, aspect** and road network   density (Hyandye et al. (2015).’ * what do you mean   by soil cover, do you mean vegetation cover? ** do you mean slope aspect?

Answer

The suggested change was not made,   because we consider that it loses the meaning of the sentence. However, it   was but specified: "soil cover" and "slope aspect".

Line 72

You can   delete ‘Similarly’ at the start of this sentence. Sentence can start ‘Dong et   al. …’

Answer

The suggested changes were made. The   text was edited with change control.

Lines   73-77

Change   sentence to “Qu et al. (2015) analyzed trends in NDVI between 1982 and 2011   for wetlands in China. They were able to correlate them with climatic   conditions and indicators on human activity, and developed a multiple   regression model that determined the response of vegetation to climatic   factors.’

Answer

The suggested changes were made. The   text was edited with change control.

Line 78

Change   ‘authors highlighted’ to ‘authors have highlighted’.

In   addition this sentence (Lines 78-80) should be moved to the start of the   paragraph, therefore insert at Line 65. The change the sentence which starts   at Line 65 in the current version of the paper to ‘In   the context of this research, NDVI has been uses to detect vegetation cover   in wetlands and determine the extent of wetland areas (Beck et al., 1990).

Answer

The suggested changes were made. The   text was edited with change control.

Line 81

Start a   new paragraph with the sentence beginning ‘Today,   there are…’

Answer

The suggested changes were made. The   text was edited with change control.

Lines 83

Change   ‘evaluating the land use change and the variation of the’ to ‘evaluation of land use changes and variations in’

Answer

The suggested changes were made. The   text was edited with change control.

Line 88

Change   ‘change’ to ‘changes’ and change ‘may be due’ to ‘may   also be due’

Answer

The suggested changes were made. The   text was edited with change control.

Line 90

Change   ‘Li et al. (2018), evaluated’ to ‘Li et al. (2018), have   evaluated’

Answer

The suggested changes were made. The   text was edited with change control.

Line   93-94

The last   sentence in this paragraph provides an accuracy value – 91% - which must be   specific to one study. I assume from Li et al. (2018). If that is so, I would   suggest changing the wording on Lines 90-94 as follows: ‘Li et al. (2018)   evaluated the dynamics and loss of marsh and meadow   vegetation in wetland due to hydrological variations using NDVI data derived   from Landsat TM, ETM+ and OLI sensors, and   generated accuracies up to 91%.’

Answer

The suggested changes were made. The text   was edited with change control.

Line 97

Change   ‘Landsat satellite images (sensors TM and OLI),’ to Landsat   TM and OLI images’

Answer

The suggested changes were made. The   text was edited with change control.

Line 99

Change   ‘and a groundwater…’ to ‘and data from a groundwater…’

Answer

The suggested changes were made. The   text was edited with change control.

Section 2.1

Line   106-7

I think   I know what confined to the ground means, and if I am correct I think the   following change needs to be made so the sentence ends as follows ‘…free   column confined in the ground’.

Answer

The suggested changes were made. The   text was edited with change control.

Line   107-109

The   sentence would be better if rephrased. ‘The area is fed by phreatic   discharges from the MNT-aquifer (Keddy, 2008; Zedler et al., 2008).

Answer

The suggested change was not made,   because we consider that it loses the ecological sense of the   characterization of the ecosystem.

Footnote   1

Line 2:   change ‘visible and Infra-Red’ to ‘visible and near   infra-red (NIR)’

Line 2:   change ‘Electromagnetic Spectrum’ to ‘electromagnetic   spectrum’

Line 3:   normally is called Enhanced Thematic Mapper (you do not need to include Plus)

Lines   3-4: change ‘visible, Near Infra-Red (NIR), shortwave and Thermal Infra-Red   (TIR)’ to ‘visible, NIR, shortwave and thermal   infra-red’’

Line 5:   change infrared (twice) on this line to infra-red, to be consistent.

Answer

The suggested changes were made. The   text was edited with change control.

Line   109-110

Change   ‘conformed by azonal vegetation (dependence on the phreatic levels) to ‘confirmed by the phreatically-influenced azonal   vegetation’

Answer

The suggested changes were made. The   text was edited with change control.

Line 110

Change  ‘transition’ to ‘transitional’.  Also Line 328

Answer

The suggested changes were made. The   text was edited with change control.

Line 111

Change   ‘hygrophilics’ to ‘hygrophilic species’

Answer

The suggested changes were made. The   text was edited with change control.

Line 114

Change   ‘ransition’ to ‘transitional’ and change   ‘humidity available in the soil’ to ‘soil moisture   levels’

Answer

The suggested changes were made. The   text was edited with change control.

Lines   115-117

Make a   new sentence after my suggested change to ‘…soil moisture levels’. The next   sentence might read as follows ‘The wetland contracted and expanded according   to annual rainfall during the time period analyzed; it was noted to expand in   wet year (e.g., during the year with the highest wet season rainfall, 2002)   and contract in dry years (e.g., 1991, Figure 2).’

Answer

The suggested changes were made. The   text was edited with change control.

Line 118

Change   ‘presented’ to ‘experiences’

Answer

The suggested changes were made. The   text was edited with change control.

Line 120

Change   ‘the historical series,’ to ‘records’

Answer

The suggested changes were made. The   text was edited with change control.

Lines   121-122

Delete   ‘it was determined that’

Answer

The suggested changes were made. The   text was edited with change control.

Lines   122-123

Change   ‘concentrating on the period between’ to ‘most of   which occurs’

Answer

The suggested changes were made. The   text was edited with change control.

Line 123

Change   ‘reaching’ to ‘of’

Answer

The suggested changes were made. The   text was edited with change control.

Line 124

Delete   ‘on average due to extreme events’

Answer

The suggested changes were made. The   text was edited with change control.

Line   124-5

Change   the sentence ‘The minimum precipitation ….’ to ‘The   lowest monthly precipitation totals (0.00 mm +/- 0.00 mm) occur between   September and November.’

Answer

The suggested changes were made. The   text was edited with change control.

Line 126

Change   ‘historical average temperature is’ to ‘historical   average annual temperature was’

Answer

The suggested changes were made. The   text was edited with change control.

Lines   126-127

Start a   new sentence ‘The coldest month is…rest of sentence correct’

Answer

The suggested changes were made. The   text was edited with change control.

Line 129

This   sentence should be part of the paragraph above. Change ‘average of   evapotranspiration’ to ‘average evapotranspiration’

Answer

The suggested changes were made. The   text was edited with change control.

Section 2.2

Line 133

Change   ‘with’ to ‘by’

Answer

The suggested changes were made. The   text was edited with change control.

Lines   134-5

Change   ‘two satellite images’ to ‘two Landsat TM or OLI*   images’ the wording in parentheses to (dry season   – September, and wet season – March). Omit ‘obtained from the Landsat   Platform (TM and OLI sensors).’ *In Footnote 1 you   wrote that you used TM, ETM+ and OLI imagery, but this sentence you do not   mention ETM+. Please make sure that you are consistent on whichever is   correct throughout the paper.

Answer

The suggested changes were made. The   text was edited with change control.

Lines   136-138

It is   unclear to me whether the images had already had radiometric and atmospheric   correction before you acquired them from NASA/USGS or whether you carried out   these two corrections. If it is the first, you need to mention that, and if   it is the second case, you need to say that.

Answer

This process was developed in the   context of this investigation. It was detailed in the text, which corresponds   to an activity carried out by the authors: “then we”.

Lines   141-143

After ‘…   Pan et al., 2018)’ start a new sentence which will read something like ‘The use of this NDVI threshold is supported by research   in other hyper-arid areas in Chile (Castro et al., 2014; Montalva et al.,   2015) and was complemented by visual analysis of true colour composites.’

Answer

The suggested changes were made. The   text was edited with change control.

Lines   144-145

Change   ‘wet, dry and annual period’ to ‘the wet and dry   season images and the ‘annual’ image, which is the average of the wet and dry   season images’.

Answer

The suggested changes were made. The   text was edited with change control.

Section 3.2

Lines   237-43

This can   be written more succinctly as ‘The piezometers and   observation wells in the groundwater monitoring network have a number of   known errors due to altitudinal variations (CEA, 2015). Measurements from   eight piezometers and obersvation wells were used to the construct the   prediction model of the land surface covered by vegetation; all erroneous   measurements were excluded.’

Answer

The suggested changes were made. The   text was edited with change control.

Line   247-8

‘average’   should be ‘averages’ as it refers to both precipitation and   evapotranspiration. Also note this change needs to be made on Lines 263 and   278.

This   line can be changed from ‘…evapotranspiration and the depth levels of the   water table from the piezometers…’ to …evapotranspiration   and the water table levels from piezometers…  A similar change also needs to be made on   Lines 264 and 278-9.

Answer

The suggested changes were made. The   text was edited with change control.

Line 250

Change   ‘…well TP-2 [Equation (1)], was the results of 97 iterations performed to   determine the best fit.’ To ‘…well TP-2. Ninety-seven   iterations were performed to determine the best fit to Equation (1).

Answer

The suggested changes were made. The   text was edited with change control.

Line 254

Should   the description for TP2 be ‘the level for the observation well   TP-2 for the wet period (m)’ so that is in the same format as the description   for the piezometers in line 253? Also on Line 284.

Answer

The suggested changes were made. The   text was edited with change control.

Line 258

I think   ‘strong correlations’ would recognised by more   readers that ‘good settings’. Refer also to Lines 272-3 and Line 287.

Answer

The suggested changes were made. The   text was edited with change control.

Line   265-6

Change   ‘…TPZ-12 [Equation (2)], after 96 iterations, were performed to determine the   best fit.’ To ‘…TPZ-12. Ninety-six iterations were   performed to determine the best fit to Equation (2).

Answer

The suggested changes were made. The   text was edited with change control.

Line   269-270

I note   that you use two formats for naming the piezometers and the observation well,   e.g. TP2 and TP-2. You need to choose one format and stick to it,   as you TPused in the   equations, I might suggest that that format is used in the text as   well as it will be easier to change.

Answer

The suggested changes were made. The   text was edited with change control.

Line   279-80

Change   ‘…well, TP-2 (Equation (1)), after 99 iterations, were performed to determine   the best fit.’ To ‘…well TP-2. Ninety-nine iterations   were performed to determine the best fit to Equation (3).

Answer

The suggested changes were made. The   text was edited with change control.

Line 291

Change   ‘minimum’ to minimal’.

a) But I have a question. In what ways were these   changes minimal and how did you compare them? You need to add a sentence that   tells the reader this information.

b) In addition, I do not understand what ‘softer’   changes means. Can you explain in more scientific terms? Also ‘sharp’   changes, e.g. Line 313. I wonder if by changes you mean boundaries between   vegetated and non-vegetated surfaces. In which case I might suggest terms   like gradational boundaries for softer. Sharp boundaries would make sense so   would not need changing.

Answer

The suggested changes were made. The   text was edited with change control.

a) The explanation was added in the   text: “and was obtained by calculating the difference between both”.

b) It was replaced by: “presented less   variability, compared to the dry and wet model”. In addition it was added:   “The annual model also showed an average difference between the estimated   values with vegetated areas mapped from satellite images and modeled data of   0.18 ± 6.89 ha, and the major differences between the estimated and modeled   data occurred when extreme variations in the surface covered with vegetation   were observed, product of the events of precipitation and   evapotranspiration”.

Line 298-99

Change   ‘… data, according to the predictive models…’ to data   from the predictive models…’

Answer

The suggested changes were made. The   text was edited with change control.

Line 303

Change   ‘setting’ to ‘correlation’. Also Line 308, Line   314

I have a   question regarding the term ‘values estimated and modeled’.

I think you are using the term estimated incorrectly,   do you means the areas you measured from the satellite images? If so, I think   it would be better to replace estimated values with vegetated areas mapped   from satellite images and modelled areas. This might apply elsewhere in the   manuscript, e.g. Lines 307-8, 311-12, 314-5

Answer

It was replaced where it was required.   In some places it was left as "estimated", to avoid redundancy in   the paragraph.

Line 316

Change   to ‘but worse that the wet season model (Figure 8c).’

Answer

The suggested changes were made. The   text was edited with change control.

Section 4

Line 318

Change   ‘difference observed in the descents and depths of the water table’ to ‘differences observed in the changes in water table   depths’

Answer

The suggested changes were made. The   text was edited with change control.

Line 320

Change   ‘permeability’ to ‘permeabilities’

Answer

The suggested changes were made. The   text was edited with change control.

Line 323

Change   ‘product of’ to ‘produced by’

Answer

The suggested changes were made. The   text was edited with change control.

Line 325

Can you please explain differentiated decrease?

Answer

At the water table, it descends   differently at each measurement point. This is due to soil variations (such   as permeability), fissures, or others.

Added a footnote in the text: “This is   due to soil variations (such as permeability), fissures, or others”.

Line 326

Change   ‘increase the aeration’ to ‘increase in the aeration’

Answer

The suggested changes were made. The   text was edited with change control.

Line 327

Change   ‘humidity’ to ‘soil moisture’

Answer

The suggested changes were made. The   text was edited with change control.

Line 330

Change’   the decrease in the water table are’ to ‘lowering of   the water table is’

Answer

The suggested changes were made. The   text was edited with change control.

Line 332

Change   ‘…Tilopozo, as a function if time, in the period, 1985-2002, showed vaiable   behaviour, with intra-…’ to ‘…Tilopozo between 1985   and 2002 showed both intra-…’

Answer

The suggested changes were made. The   text was edited with change control.

Line 335

Change   ‘…satellite imagery and NDVI index…’ to ‘…NDVI…’.

Answer

The suggested changes were made. The   text was edited with change control.

Lines   399-340

Change   ‘…the phenological stage in which they are located.’ to ‘…their phenological stage.’

Answer

The suggested changes were made. The   text was edited with change control.

Lines   340-8

There   are two long and complex sentence which could be better written as follows. ‘One of the most important and extensive plants in the   salt meadows of Tilopozo is S.   americanus.  However, Langley and   Megonigal (2002) indicated that determining the phenological stage of S. americanus using NDVI is very   complex because the biomass is brown and this generate very low NDVI values   which can be interpreted as bare soil. In addition as S. americanus plants increase in height their leaf areas   decreases, which impacts NDVI in an counterintuitive manner, because higher   biomass leads to lower NDVI values (Kearney et al., 2009).

Answer

The suggested changes were made. The   text was edited with change control.

Lines   353-9

Rewrite   long sentence as ‘Meanwhile, du Plessis (1999)   pointed out that statistical relationships between vegetation cover and   precipitation in arid-zone grasslands are deficient because there are factors   that determine vegetation behavior that cannot be measured from satellite   imagery such as slope, soil moisture and water table depth. Sorrell al.   (2012) also identified water table depth as a factor that determines   hygrophyte species cover.

Answer

The suggested changes were made. The   text was edited with change control.

Line 364

This   could be better written as ‘…map based in NDVI and obtained accuracies   between…’

Answer

The suggested changes were made. The   text was edited with change control.

Line 365

Start   sentence as follows ‘In arid-zone wetlands in Spain,   …’

Answer

The suggested changes were made. The   text was edited with change control.

Line   366-7

Change   ‘…evaluated the wetlands of China from an NDVI value of 0.13, using the same   values as in the present investigation.’ to ‘…evaluated   Chinese wetlands using an NDVI threshold of 0.13, the same value used in this   investigation.’

Answer

The suggested changes were made. The   text was edited with change control.

Line 370

Do you mean salt meadows?

Answer

No. It refers to the instrument that   has been used in similar situations.

It was added in the text: “in similar   situations”.

Line 372

Delete   ‘as is the case for Tilopozo’

Answer

The suggested changes were made. The   text was edited with change control.

Line 372

You use   the term ‘higlights’ here, in previous sentence you used ‘highlight’ and in   some places you have used ‘indicated’ the equivalent of which would be   ‘highlighted’. I think it would be best to decide on one tense (present or   past) and use it throughout the paper.

Answer

The suggested changes were made. The   text was edited with change control.

Lines   372-6

These   sentences could be better written as ‘Purevdorj et al. (188) highlights that   SAVI is a better indicator of vegetation cover at low densities than NDVI.   However, it is necessary to have soil correction factors to calculate SAVE   which restricts its application (Navarro et al., 2006; Abaurrea, 2013).

Answer

The suggested changes were made. The   text was edited with change control.

Lines   378-9

Change   ‘’… , the agricultural and livestock activity…’ to ‘… , cultivation and grazing…’

Answer

The suggested changes were made. The   text was edited with change control.

Lines   380-2

This   sentence is too long and has too many clauses. So I suggest that a new   sentence is added after ‘… salt meadow’. This could read, ‘Burning is not   formally registered* but was observed by the authors during field campaigns   (Figure 9) and is likely to reduce NDVI values.

Answer

The suggested changes were made. The   text was edited with change control.

Lines   382-389

The   material from sentence ‘NDVI had been used… … (Tian et al., 2016)’ to the end   of this paragraph (Line 389) needs to be moved earlier in Section 4 where you   discuss other research using NDVI.

Answer

The suggested changes were made. The   text was edited with change control.

Line 391

‘… was   lower, …’ is a comparator, therefore it has to be ‘… was   lower than the human actions discussed above. This may be …’/

Change   ‘hygrophila’ to ‘hygrophilous’

Answer

The suggested changes were made. The   text was edited with change control.

Line 392

Change   ‘…  conditions, depending… ’ to ‘… conditions, but rather depends primarily …’

Answer

The suggested changes were made. The   text was edited with change control.

Line 401

Change   ‘presented models (we, dry and annual periods)’ to ‘models   developed in this research for the wet dry and annual periods’.

Answer

The suggested changes were made. The   text was edited with change control.

Lines   402-3

a) I do not understand what ‘corroborated on the obtention of the   best R2 and R2-adjustment, as well as statistical significance thereof’   means.

b) Nor do I understand what ‘the characteristics’ referred to in   the next sentence are’

Answer

a) The performance of the models is corroborated   with 3 statistics: R2, R2-adjusted, and the statistical significance of   these.

b) It refers to the three statistical   indicators: R2, R2-adjusted and statistical significance of the model.

Line 408

Add   ‘from’ so it reads ‘estimated from satellite images’

Change   ‘…exceed 1 ha, with a total of 90 average surface…’ to ‘exceed 1 ha within all 90 surfaces developed…’

Answer

The suggested changes were made. The   text was edited with change control.

Line   410-11

Change   ‘..abrupt changes in the surface…’ to ‘…abrupt   increases and decreases in the surface…’ and delete ‘either increasing or   decreasing’ at the end of the sentence.

Answer

The suggested changes were made. The   text was edited with change control.

Line   412-3

Change   ‘…events, such as what happened in 2002, where a precipitation of 40.5 mm was   recorded in 24 h, while the normal annual rainfall is 20 mm year-1’   to change ‘events, such as the 40.5 mm of   precipitation recorded in 24 hours in 2002, which can be compared to the mean   annual precipitation of 20mm.’

Answer

The suggested changes were made. The   text was edited with change control.

Line   415-6

Delete  ‘…, as has been reported in other sectors…’

Answer

The suggested changes were made. The   text was edited with change control.

Line 416

Change   ‘…abnormal…’ to ‘…such…’

Answer

The suggested changes were made. The   text was edited with change control.

Line 417

Change   ‘…, which can generate under- or over-estimates in the modelled values…’ to   ‘… and can lead to under- and overestimates in   predicted values…’

Answer

The suggested changes were made. The   text was edited with change control.

Line 428

Should   relictus start with an upper case R? and I think the comma between relictus   and National is not required.

Answer

The suggested changes were made. The   text was edited with change control.

Line 430

‘water   mirror’ is not a recognisable scientific term in English. Also line 432

Answer

The suggested changes were made. The   text was edited with change control.

Was replaced by: “free water column”.

Line 434

Do you   mean prediction tools rather than prevention tools

Answer

Was replaced by: “prediction”.

Additionally, it was highlighted that   it can be used as a conservation tool: "being useful as a conservation   tool".

Lines   433-437

I think   this sentence could be rewritten as ‘The development of prediction models   allows an understanding the reasons for the fluctuations in the Tilopozo salt   meadow(s?) to be developed and the influences of human actions and climate   change in causing these changes. Climate change may be particularly important   as it also affects water availability in highland regions (Espinoza and   Romero, 2015).

Answer

The suggested changes were made. The   text was edited with change control.

Line   437-440

I do not understand the point you are trying to make in the last   sentence.

Answer

Was replaced by: “The prediction models   will allow evaluating the state of the salt meadows and will therefore allow   necessary measures to be taken in the event of a substantial change in the   surface covered with vegetation, being useful as a conservation tool”.

Section 5

Line   443-4

Change   the final clause of the sentence to ‘… , which   equates to an average annual loss of 1.08 ha year-1.’ Also   combine this sentence into the next paragraph, and use it to start the next   paragraph.

Answer

The suggested changes were made. The   text was edited with change control.

Line 445

Change   ‘This was’ to ‘These changes were’ and ‘the   variation of the water table.’ to ‘water table   variations.’

Answer

The suggested changes were made. The   text was edited with change control.

Line 448

Meadows   should start with a lower case m

Answer

The suggested changes were made. The   text was edited with change control.

Lines   449-50

Change   sentence to ‘The behaviour of the vegetation cover   was less dependent on meteorological conditions.’

Answer

The suggested changes were made. The   text was edited with change control.

Line 451

Change   ‘vegetal’ to ‘vegetation’

Answer

The suggested changes were made. The   text was edited with change control.

Line 445

Change   ‘and the estimates information obtained by remote sensing’ to ‘vegetation area estimates from satellite data.’

Answer

The suggested changes were made. The   text was edited with change control.

Thanks for the feedback! They have helped to improve this publication.
